# Mass Spectrometric Quantification of the Antimicrobial Peptide Pep19-2.5 with Stable Isotope Labeling and Acidic Hydrolysis

**DOI:** 10.3390/pharmaceutics13091342

**Published:** 2021-08-27

**Authors:** Sabrina Wohlfart, Michael Kilian, Philip Storck, Thomas Gutsmann, Klaus Brandenburg, Walter Mier

**Affiliations:** 1Department of Nuclear Medicine, Heidelberg University Hospital, 69120 Heidelberg, Germany; sabrina.wohlfart@med.uni-heidelberg.de (S.W.); m.kilian@dkfz-heidelberg.de (M.K.); philip.storck@med.uni-heidelberg.de (P.S.); 2Division of Biophysics Research Center Borstel, Leibniz-Center for Medicine and Bioscience, 23845 Borstel, Germany; tgutsmann@fz-borstel.de; 3Brandenburg Antiinfektiva GmbH, Parkallee 10b, 23845 Borstel, Germany; kbrandenburg@fz-borstel.de

**Keywords:** sepsis, antimicrobial peptides, SALP, peptide labeling, stable isotope labeling, mass spectrometry

## Abstract

Sepsis is the number one cause of death in intensive care units. This life-threatening condition is caused by bacterial infections and triggered by endotoxins of Gram-negative bacteria that leads to an overreaction of the immune system. The synthetic anti-lipopolysaccharide peptide Pep19-2.5 is a promising candidate for the treatment of sepsis as it binds sepsis-inducing lipopolysaccharides and thus prevents initiation of septic shock. For clinical evaluation precise quantification of the peptide in blood and tissue is required. As the peptide is not extractable from biological samples by commonly used methods there is a need for a new analysis method that does not rely on extraction of the peptide. In order to quantify the peptide by mass spectrometry, the peptide was synthesized containing ^13^C_9_,^15^N_1_-labeled phenylalanine residues. This modification offers high stability during acidic hydrolysis. Following acidic hydrolysis of the samples, the concentration of ^13^C_9_,^15^N_1_-labeled phenylalanine determined by LC-MS could be unambiguously correlated to the content of Pep19-2.5. Further experiments validated the accuracy of the data. Moreover, the quantification of Pep19-2.5 in different tissues (as studied in Wistar rats) was shown to provide comparable results to the results obtained with radioactively-labeled (^14^C) Pep19-2.5- Radioactive labeling is considered as the gold standard for quantification of compounds that refrain from reliable extraction methods. This novel method represents a valuable procedure for the determination of Pep19-2.5 and sticky peptides with unpredictable extraction properties in general.

## 1. Introduction

Bacterial infections cause approximately 8–9 million of deaths each year worldwide. In most cases, bacterial infections can be treated by the use of antibiotics. Unfortunately, the increasing prevalence of resistant germs (the best-known being MRSA), augments the number of systemic infections eventually resulting in sepsis [1,2]. The burden of sepsis is the number one cause of death in intensive care units and estimated to affect 15% of deaths worldwide [3]. On the contrary, the number of new antibiotics approved by the authorities (e.g., FDA, BfArM) is continuously decreasing [4,5]. On that score, a gap in supplies of antibiotics will arise in the future [6]. Closing this gap with drugs providing the required pathogen-specificity can be achieved with drugs that specifically reduce the toxicological effects of bacteria.

Antimicrobial peptides (AMP) constitute a promising therapeutic option [7]. In contrast to standard antibiotics that interact directly with bacteria by growth inhibition the reproduction or cytotoxicity, AMP bind lipopolysaccharides (LPS), compounds specifically occurring in the outer membrane of Gram-negative bacteria. LPS are essential compounds of the bacterial cell wall and nontoxic. When released from the cell wall into the human blood system, LPS constitute a major sepsis induction factor. LPS bind to CD14, a surface protein mainly located on macrophages and monocytes. CD14 forms a complex with individual receptors such as TLR4, MD-2, and other surface proteins. The CD14-bound TLR4 triggers intracellular signal pathways, which activate immune cells. These cells release pro-inflammatory cytokines which activate the innate immune system provoking an inflammatory response (Figure 1) [8].

Lysis of bacteria leads to increased release of LPS followed by an increased activation of the immune system. This eventually leads to sepsis, a detrimental host response to infection. Over-activation of the immune system can lead to coagulation disorders and organ dysfunction. Currently, cardiorespiratory resuscitation and early intravenous antibiotic therapy represent current standard therapy of sepsis [9].

In order to neutralize LPS, Brandenburg et al. developed anti-LPS-peptides (synthetic anti-lipopolysaccharide peptide = SALP) [10]. The peptides are characterized by their high specific binding to LPS and a low toxicity under physiological conditions and are thought to prevent sepsis (Figure 1). Sequences of the SALPs were originally based on the *Limulus* anti-LPS factor, but the subsequent rational design resulted in the selection of the novel patent-protected Pep19-2.5 (also known as Aspidasept), being the most promising candidate out of a large number of sequence variations. It shows high binding affinity for free LPS, high endotoxin neutralization capacity in vitro and antiseptic and anti-inflammatory effects in mouse models. In contrast to alternative approaches, the peptide shows an excellently balanced anti-endotoxic versus antibacterial ratio. Furthermore, experiments show the peptide to inhibit cytokine production in human immune cells [11]. In addition, Pep19-2.5 binds and neutralizes bacterial pathogenic factors from Gram-positive bacteria [12]. It shows that the peptide is able to neutralize free and membrane- bound toxins [13]. The highly specific binding to LPS is probably achieved by the amphiphilic character of Pep19-2.5. The predominantly positive charged moiety of the peptide, interacting with the negative charged moiety of LPS, is supplemented by hydrophobic C-terminus of the peptide resulting in a surface that matches the complementary structural elements of LPS [14].

The peptide is not extractable from biological samples, so another method for analysis is necessary. One standard procedure is the organic solvent extraction of serum and following LC-MS analysis of the supernatant. For this, different solvents like methanol, acetone, chloroform or acetonitrile are most commonly used. In all cases the pH-value is important for the extraction efficiency [15]. To increase the efficiency, it might be possible to add a chelating agent like EDTA or use non-ionic hydrophilic polymers. Another possibility is the solid phase extraction of the peptide/drug with a polymeric sorbent and a following drying process prior to analysis [16]. For the analysis of organ distribution, a useful tool is MALDI-TOF analysis of organ sections [17]. An additional way for quantification without extraction and for organ distribution, is the use of a radioactive labeled compound and the measurement of radioactivity in the different samples [18]. With this method an indirect quantification of the amount of peptide is correlated.

As of now there was no analytical method for blood and tissue samples of the antimicrobial peptide Pep19-2.5 except for methods that involve radioactive labeling. Here, we report a novel method for the quantification of Pep19-2.5 by implementing marker amino acids. The marker amino acids are labeled with stable isotopes. Here, ^13^C_9_,^15^N_1_-labeled phenylalanine was used, which had no effect on the antimicrobial activity. The quantification of the peptide in biological samples (e.g., serum) is achieved by acidic serum hydrolysis without requiring extensive handling. The amount of ^13^C_9_,^15^N_1_-labeled phenylalanine quantified by LC-MS can be unambiguously correlated to the dosed amount of Pep19-2.5. The great advantage is that this method can be used for serum and for analysis of all other biological samples (e.g., organ distributions). An overview of this method is given in Figure 2.

## 2. Materials and Methods

### 2.1. Peptide Synthesis

^13^C_9_,^15^N_1_-labeled phenylalanine (Sigma-Aldrich GmbH, Taufkirchen, Germany) was Fmoc-protected using standard procedures [19]. Subsequently, the peptide synthesis of the Pep19-2.5 derivative containing four ^13^C_9_,^15^N_1_-labeled phenylalanine residues (analyte, Figure 3) was carried out by standard manual Fmoc/tBu solid-phase peptide synthesis on Fmoc-Rink amide resin (TentaGel R RAM, Rapp Polymere GmbH, Tübingen, Germany) [20]. The Fmoc-protected L-amino acids were purchased from ORPEGEN Peptide Chemicals GmbH (Heidelberg, Germany). For coupling of the isotope-labeled amino acid, 2 equivalents and a coupling time of 2.5 h was used.

The synthesis of deuterated Pep19-2.5 was carried out on an automated synthesizer (433A, Applied Biosystems, Waltham, MA, USA) on Fmoc-Rink amide resin. For the synthesis of the Pep19-25 derivative containing 32 deuterium atoms, eightfold deuterated Fmoc-protected phenylalanine (Sigma Aldrich, GmbH, Taufkirchen, Germany) was used and applied in excess of two equivalents. Pep19-2.5 labeled with ^14^C was obtained from Rainer Bartels (Leibniz-Zentrum für Medizin und Biowissenschaften, Borstel, Germany).

### 2.2. Acidic Hydrolysis

For acidic hydrolysis 10 µL ^13^C_6_(ring) phenylalanine (1 mg/mL stock) were added as internal standard to 200 µL water containing ^13^C_9_,^15^N_1_ phenylalanine (for calibration curve) or human serum (from voluntary donors) containing Pep19-2.5 with four ^13^C_9_,^15^N_1_ phenylalanine residues. After addition of 270 µL hydrochloric acid (10 M), the samples were hydrolyzed at 120 °C for 24 h. Subsequently, the solvent was evaporated in a vacuum evaporator (HETOVAC) at 40 °C for 4 h and the pellet was resuspended in water to achieve the applied concentration.

### 2.3. HPLC Analysis

HPLC analysis was performed using an Agilent 1100 system with an UV-detection at 214 nm. As stationary phase a Chromolith^®^ Performance RP-18e 100–3 mm column (Merck KGaA, Darmstadt, Germany) was used. A 5 min linear gradient from 100% water (+0.05% trifluoroacetic acid (TFA)) to 100% acetonitrile (+0.05% TFA) was applied.

### 2.4. LC-MS Analysis

LC-MS analysis was performed using an ESI-Orbitrap mass spectrometer (Exactive, Thermo Fisher Scientific, Waltham, MA, USA) coupled to a 1200 series HPLC (Chromolith^®^ Performance RP-18e 100–4.6 mm, Agilent Technologies, Santa Clara, CA, USA). A 20 min linear gradient from 100% water (+0.05% TFA) to 100% acetonitrile (+0.05% TFA) was used. The mass range was between 100 and 400 *m*/*z*. For MS peak integration, the software implemented in the mass analyzer was used (Qual Browser, Thermo Scientific, Waltham, MA, USA).

### 2.5. LC-MS/MS Analysis

The LC-MS/MS system (Thermo Fisher Scientific, Waltham, MA, USA) consisted of a P4000 LC pump, and a triple-stage quadrupole mass spectrometer (Thermo TSQ 7000) equipped with electrospray ionization (API-2 ion source) at 4.5 kV. For chromatographic separation, a Phenomenex Synergy Hydro RP column (4 μm, 2.1 × 150 mm) with an integrated pre-column was used at 50 °C. The eluent consisted of ammonium acetate (5 mM including 0.1% aqueous formic acid, 2.5% acetonitrile, and 2.5% methanol) (A) and acetonitrile/methanol (50/50) (B). The flow rate was 0.5 mL/min and was introduced without splitting into the ESI source. The gradient started at 100% A. Within 3 min, the ratio was changed linear to 60% A/40% B and subsequently within 0.5 min changed linear to 6% A/94% B and kept stable until 6 min. Within the next 0.5 min, the system returned to starting conditions, and another 2.5 min were given for equilibration. The injection volume was 50 μL. The TSQ 7000 was tuned automatically to phenylalanine and the internal standards using the Xcalibur 1.4 system software and standard optimization procedures. Multiple reaction monitoring (MRM) analysis was performed using argon as collision gas at 2.0 mTorr for collision-induced dissociation (CID), and the MS/MS transitions monitored in the positive ion mode were *m*/*z* 166.1→*m*/*z* 120.1 for Phe, *m*/*z* 171.1→*m*/*z* 125.1 for Phe-d5, and *m*/*z* 174.1→*m*/*z* 128.1 each at 18 V.

### 2.6. Biodisdribution of ^14^C-Labeled Pep19-2.5

Isotope labeled compounds represent the current gold standard for organ distribution studies. Animal treatment was carried out at the Department of Nuclear Medicine, University Hospital Heidelberg, in accordance with institutional guidelines and the German animal welfare act. NMRI mice (Janvier Labs, Le Genest-Saint-Isle, France) were anesthetized with isoflurane. 100 µL of the radioactive peptide (corresponding to approximately 70,000 cpm) supplemented with cold peptide at a concentration of 1 mg/mL (in 0.9% NaCl) were injected intravenously into the tail vain of the NMRI mice and the organs were dissected at the defined times after sacrificing the mice with CO_2_. The following organs were dissected: blood, heart, lung, liver, spleen, kidney, muscle, intestine, stomach, brain, femur, tail (injection site). Due to the uncommon properties of Pep19-2.5 (a substantial amount of the peptide sticks to the tail), the whole tail was taken in different sections for solubilization. Organ weights were approximated using published data and data provided by the animal supplier (Janvier Labs, Le Genest-Saint-Isle, France) [21].

Tissue samples were placed in scintillation tubes and incubated for approximately 20 h at 50–60 °C with 1 mL of solubilizer (5% *N*,*N′* dimethyldodecylamine-*N*-oxide solution (Sigma-Aldrich GmbH, Taufkirchen, Germany), 5% Tergitol type 15-S-7 (Sigma-Aldrich GmbH, Taufkirchen, Germany), 2.5% NaOH). For blood, liver, spleen, heart, lung and kidney samples, 0.1 mL of 0.1 M EDTA was added in order to avoid foam formation. For discoloration, 0.1–0.4 mL of H_2_O_2_ (30%) was carefully added and samples were incubated for 1 h at 50–60 °C. After bleaching, 10 mL of the scintillation liquid Ultima Gold (PerkinElmer Inc., Waltham, MA, USA) was added. Samples were measured in a beta-counter (1414 Wallac Liquid Scintillation Counter, PerkinElmer, Waltham, MA, USA) at room temperature.

### 2.7. Biodistribution ^2^H-Labeled Pep19-2.5

Animals were treated as described above. The in vivo experiments were approved by the Animal Welfare Board of the Governmental Office (Regierungspräsidium, Karlsruhe, Germany; permit G-127/18) and the University of Heidelberg Committee for Ethics on Laboratory Animal Experimentation and were performed in compliance with institutional guidelines, the German law for animal protection, the Directive 2010/63/EU and FELASA (Federation of European Laboratory Animal Science Associations, Ipswich, UK) guidelines. In total, 200 µL of a sterile filtered 5 mg/mL solution of the deuterated peptide Pep19-2.5 was injected into Wistar rats (Janvier Labs, Le Genest-Saint-Isle, France). The animals were sacrificed and organ samples were dissected and weighed. Depending on the experimental setup, a defined amount of the internal standard Phe-d5 (CDN Isotopes Inc., Pointe-Claire, QC, Canada) was added to the samples. For organ samples, 1 µg Phe-d5 per mg sample was added.

The samples were hydrolyzed with 6 N HCl for 24 h at 110 °C. Then, the samples were filtered with glass wool followed by sterile filtration (Millex-GV 0.22 µm, Merck). HCl was evaporated in samples of 40 µL in vacuo at room temperature for 2 h. Neutralization was achieved by addition of saturated sodium bicarbonate solution after dissolution of the samples in water. Serum samples were measured using LC-MS and organ distribution samples were measured using LC-MS/MS.

All animal trials were approved by the Animal Care and Use committees at the Regierungspräsidium Karlsruhe, Germany.

## 3. Results

In this article, a novel mass spectrometric method for the analysis of Pep19-2.5 in biological samples is described. Below, each step of the analysis method is described in detail for the ^13^C_9_,^15^N_1_ labeled compound. For the development of this method and the proof of principle in the first experiments, deuterated phenylalanine was used.

### 3.1. Selection of Amino Acid

For an accurate determination it is necessary to select an amino acid which is stable under the extreme conditions in the hydrolysis step (24 h, 120 °C). Moreover, the development of the method is facilitated if the amino acid is UV-active and more importantly in case of hydrophobic amino acids that are separated from the bulk of amino acids by RP-HPLC. Due to this, the three aromatic amino acids phenylalanine, tryptophan and tyrosine were hydrolyzed, and their stability analyzed by HPLC, as shown in Figure 4.

As seen in Figure 4 it is obvious, that only phenylalanine (Figure 4a) is sufficiently stable during the hydrolysis process. Tryptophan forms several degradation products and tyrosine is degraded to poorly soluble products as reflected by the reduction of its UV-absorption peak. Another positive aspect is that the peptide Pep19-2.5 contains four phenylalanine residues, resulting in a fourfold higher sensitivity for this amino acid after hydrolysis.

### 3.2. Normalization of ^13^C_9_,^15^N_1_ Phenylalanine to Internal Standard

First attempts using deuterated phenylalanine derivative revealed high deuterium hydrogen exchange ratios under the hydrolysis conditions. For the consistency of the measurements, it was therefore necessary to use ^13^C_6_ (ring)-labeled phenylalanine as internal standard in every sample in a defined amount of 0.05 mg/mL. With this knowledge and Equation (1), it is possible to normalize all measured ^13^C_9_,^15^N_1_ phenylalanine values and compare them.
(1)$${\mathrm{AUC}}_{\mathrm{Phe},\;\mathrm{normalized}}=\frac{\left|{\mathrm{IS}}_{\mathrm{Blank}}-{\mathrm{IS}}_{\mathrm{Sample}}\right|}{{\mathrm{IS}}_{\mathrm{Blank}}}\cdot {\mathrm{AUC}}_{\mathrm{Phe},\;\mathrm{Sample}}$$

### 3.3. Establishment of a Calibration Curve

In the subsequent step, a ^13^C_9_,^15^N_1_ phenylalanine standard curve, as shown in Figure 5, was determined for concentrations between 0.74 and 190.48 µM. The values detected were in excellent accordance with the linear plot. At lower concentrations, a reliable detection of ^13^C_9_,^15^N_1_ phenylalanine was not possible.

The linear regression resulted in:(2)$$\mathrm{concentration}\;\mathrm{Phe}\;[µM]\;=\;\frac{\mathrm{abundance}\;+\;2301}{60,499}$$

With a coefficient of determination for the linear fit of 0.9999. The calculation of the amount of Pep19-2.5 in biological samples is now possible:(3)$$\mathrm{concentration}\;\mathrm{Pep}19-2.5\;[µM]\;=\;\frac{1}{4}\;\mathrm{concentration}\;\mathrm{Phe}\;[µM]$$

### 3.4. Validation of Data

To determine the accuracy of the established method, known amounts of Pep19-2.5 (4 × ^13^C_9_,^15^N_1_ phenylalanine) were dissolved in human serum, hydrolyzed, evaporated and dissolved in water. Subsequently, the content of ^13^C_9_,^15^N_1_ phenylalanine was determined by LC-MS analysis and the quantity of Pep19-2.5 was calculated using the generated standard curve. For a clear comparison between the adjusted concentration and the determined concentration via phenylalanine standard curve, the values were plotted as shown in Figure 6.

As Figure 6 implies, the determined concentration of Pep19-2.5 (4 × ^13^C_9_,^15^N_1_ phenylalanine) is lower than the adjusted one. This was attributed to an incomplete recovery from the serum. The discrepancies between the calculated and the adjusted concentrations accounted to approximately 25% in each measurement. Based on this knowledge it is possible to introduce a correction factor and to precisely determine the amounts of Pep19-2.5 (4 × ^13^C_9_,^15^N_1_ phenylalanine) in human serum.

### 3.5. Biodistribution

The results for organ distribution of ^14^C-labeled Pep19-2.5 are shown in Figure 7. The peptide initially accumulates in the lungs resulting in high %ID/g values. The activity in the lungs decreased significantly after 15 min, whereas this case was not as strong in the other organs. Significant amounts of the peptide were located in spleen, liver, kidney and tail. %ID/g values for the spleen and intestine doubled over 4 h. Sixty minutes post injection, the activity was almost completely cleared from the blood. When considering the whole organs, the majority of the peptide was found in the liver over the entire measurement period (46 to 69%ID). A small portion was also found in the kidneys (Figure 8a,c).

Organ distribution for the deuterated Pep19-2.5 was performed in a Wistar rat 15 min post injection. The results are shown in Figure 8. Since the injected peptide solution containing the ^2^H-labeled peptide was filtered with a sterile filter, aggregation was reduced in this approach which resulted in lower accumulation and %ID/g values in the lungs. However, comparing the other organs, the approach based on quantification of the deuterated peptide resulted in similar relative %ID/g values (Figure 8a,b). When quantifying the peptide in whole organs, the majority of the peptide was found to be in the liver as consistently determined by both approaches (Figure 8c,d).

## 4. Discussion

The prevention of sepsis by Pep19-2.5 is a promising new approach, because the peptide efficiently neutralizes bacterial toxins such as LPS and could be used against whole body infections. Today, LC-MS is the standard procedure for the analysis of peptides and small proteins. In the standard analysis protocols, the analyte is separated from the matrix (e.g., the serum) and quantified using LC-MS or LC-MS/MS. In cases in which the analyte cannot be unambiguously quantified, for example because of concentrations below the detection limit or problems in matrix separation, indirect quantification by a radiolabeled derivative can be used. Most common in that case is the use of tritium and carbon-14 labeled peptides.

Until now, there was no possibility to detect the peptide Pep19-2.5 directly in animal studies or patient’s serum other than methods based on radioactively labeled derivatives. For this purpose, a novel method based on stable isotope labeling, acidic hydrolysis and LC-MS analysis to quantify the peptide in biological samples was developed. With the recorded phenylalanine standard curve, it is now feasible to calculate the amount of Pep19-2.5 in human serum and other biological samples via precise determination of ^13^C_9_,^15^N_1_ phenylalanine.

It was shown that this method can be used to determine the concentrations of Pep19-2.5 in rat organs and human serum. Until now, there was no possibility to detect the peptide Pep19-2.5 in animal studies or patient’s serum other than methods based on radioactively labeled derivatives. Furthermore, it is possible to transfer this analysis method to other peptides, which are not extractable from biological samples.

## Figures and Tables

**Figure 1 pharmaceutics-13-01342-f001:**
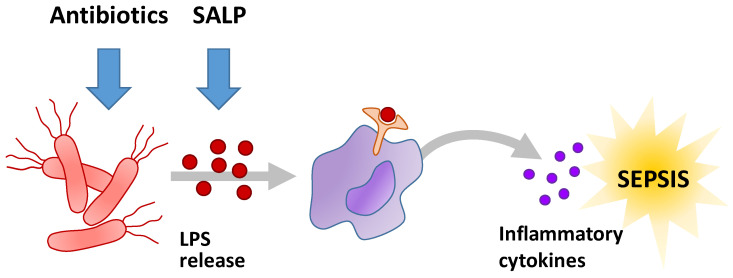
General mechanism of sepsis initiation caused bacterial infections and the target points of antibiotics and the LPS neutralizing icosapeptide Pep19-2.5.

**Figure 2 pharmaceutics-13-01342-f002:**
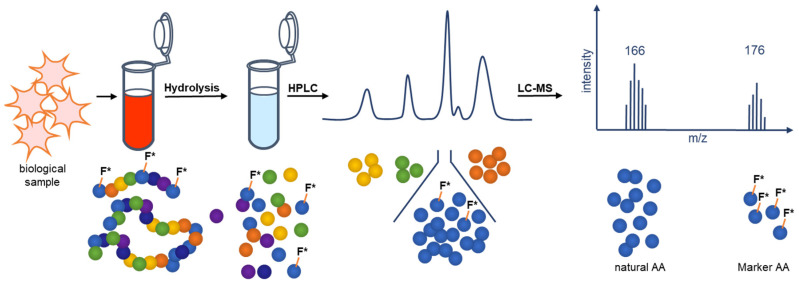
Schematic presentation of the principle of the novel mass spectrometric quantification method of Pep19-2.5 in biological samples. The quantification of Pep19-2.5 is possible after acidic hydrolysis and LC-MS analysis by implementing marker amino acids (^13^C_9_,^15^N_1_ phenylalanine).

**Figure 3 pharmaceutics-13-01342-f003:**
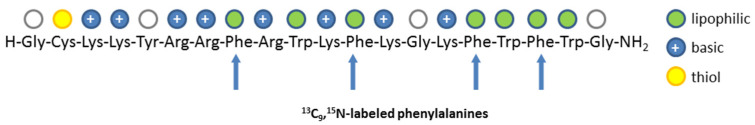
Sequence of Pep19-2.5. The arrows indicate the positions of the ^13^C_9_,^15^N_1_-labeled phenylalanine residues incorporated in the solid phase peptide synthesis.

**Figure 4 pharmaceutics-13-01342-f004:**
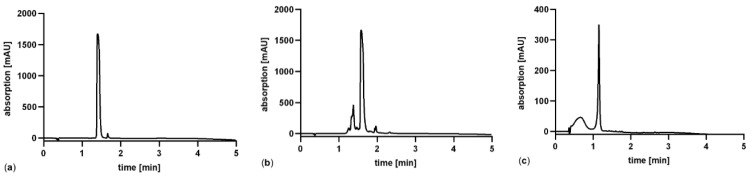
HPLC-analysis after acidic hydrolysis of aromatic amino acids (**a**) phenylalanine, (**b**) tryptophan and (**c**) tyrosine for 24 h at 120 °C. The original concentration of each amino acid prior hydrolysis was 1 mg/mL.

**Figure 5 pharmaceutics-13-01342-f005:**
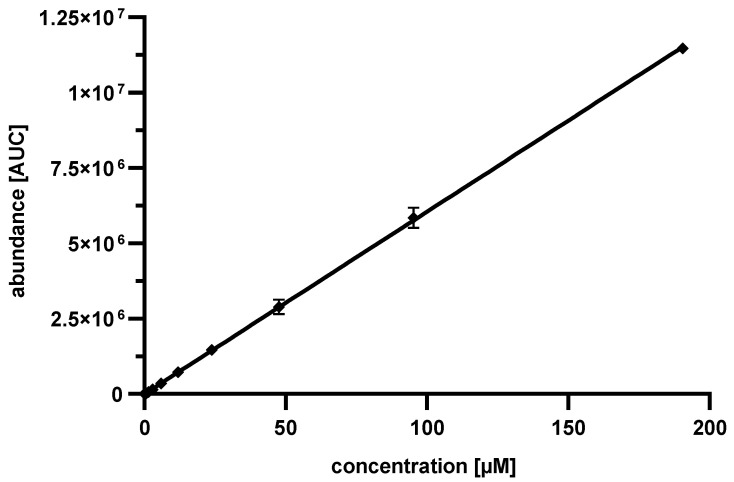
^13^C_9_,^15^N_1_ phenylalanine standard curve after acidic hydrolysis. The error was calculated by standard deviation, the number of experiments was *n* = 3, the coefficient of determination for the linear fit was 0.9999.

**Figure 6 pharmaceutics-13-01342-f006:**
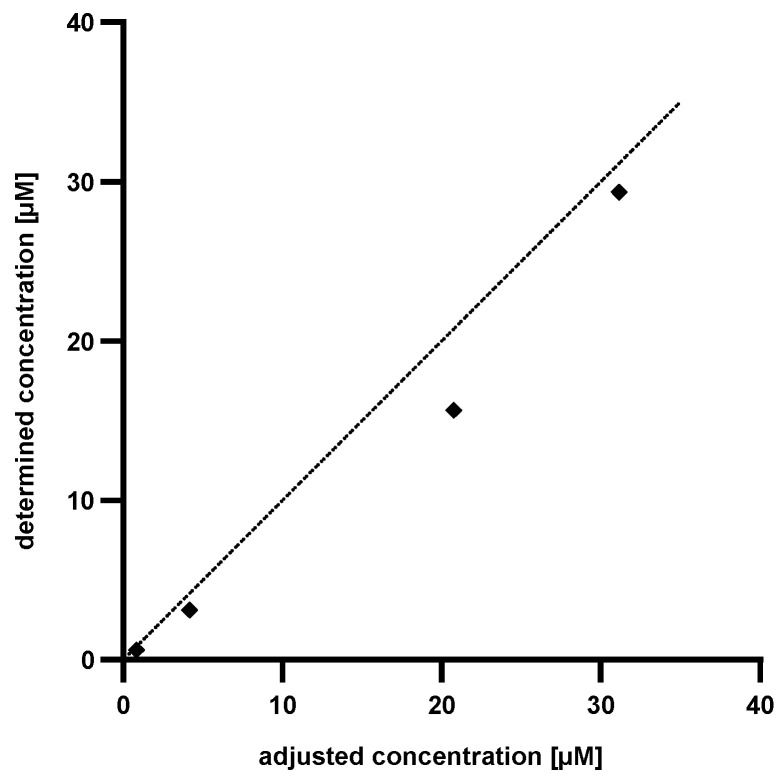
Plot of the determined concentration of Pep19-2.5 (4 × ^13^C_9_,^15^N_1_ phenylalanines) against the adjusted concentration of Pep19-2.5 (4 × ^13^C_9_,^15^N_1_ phenylalanines). The dotted line shows the calculated concentration of Pep19-2.5 (^13^C_9_,^15^N_1_ phenylalanine).

**Figure 7 pharmaceutics-13-01342-f007:**
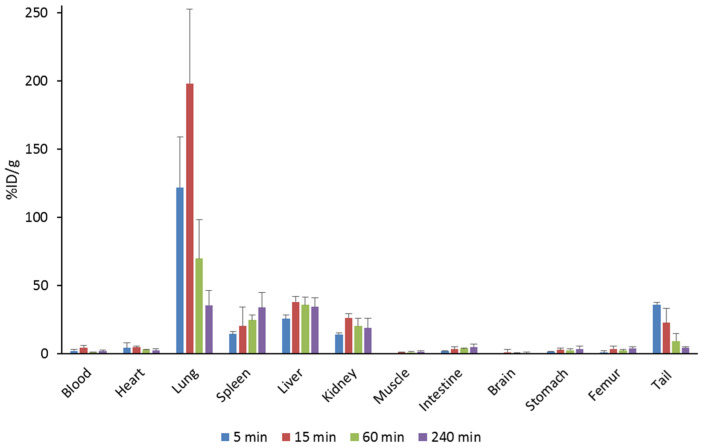
Biodistribution of ^14^C-labeled Pep19-2.5 in NMRI mice.

**Figure 8 pharmaceutics-13-01342-f008:**
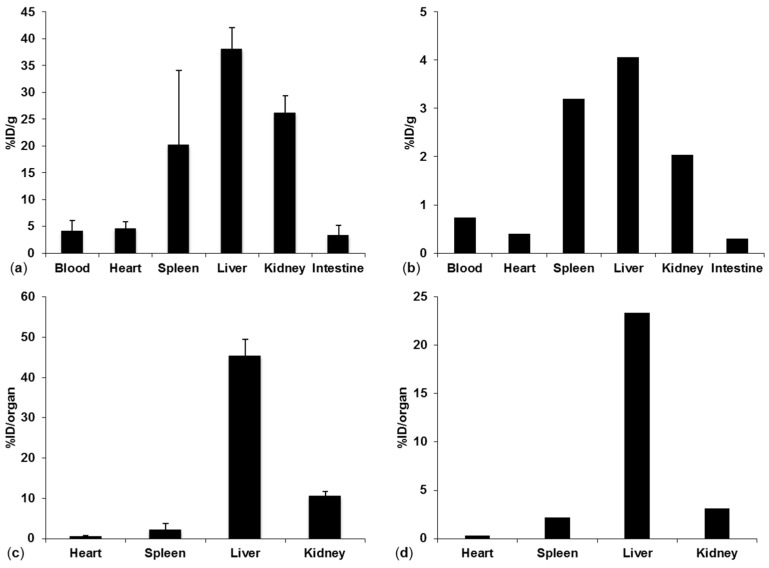
(**a**) %ID/g for ^14^C-labeled Pep19-2.5 organ distribution in NMRI mice, (**b**) %ID/g ^2^H-labeled Pep19-2.5 in a Wistar rat, (**c**) %ID/organ for ^14^C-labeled Pep19-2.5 in mice, (**d**) %ID/organ for ^2^H-labeled Pep19-2.5 in a Wistar rat 15 min post injection.

## Data Availability

Not applicable.

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
