# Peer review of "Mass Spectrometric Quantification of the Antimicrobial Peptide Pep19-2.5 with Stable Isotope Labeling and Acidic Hydrolysis"

_pharmaceutics, 2021, doi:10.3390/pharmaceutics13091342_

Round 1
Reviewer 1 Report
The submitted manuscript describes and discusses the results of an original research project carried out to mass spectrometric quantification of the antimicrobial peptide Pep19-2.5 with stable isotope labeling and acidic hydrolysis. The manuscript describes and discusses logically designed experiments and presents results that are expected to be of large interest for the scientific community. It is an interesting study with a novel approach. The paper in the whole is well designed and results sound. Nevertheless, the manuscript needs a minor revision:
(1) Abstract part should be re-written. It should be pointed more details of performed experiments.
(2) Figure 1 - should be more described in the text.
(3) Introduction part - the aim of the study should be pointed more.
(4) Figures 2 and 3 - should be removed to the experimental part.
(5) What about of Bioethical Comm. in case of real samples?
Author Response
Reviewer #1: General evaluation
The submitted manuscript describes and discusses the results of an original research project carried out to mass spectrometric quantification of the antimicrobial peptide Pep19-2.5 with stable isotope labeling and acidic hydrolysis. The manuscript describes and discusses logically designed experiments and presents results that are expected to be of large interest for the scientific community. It is an interesting study with a novel approach.
Thank you for your positive assessment of the novel analytical procedure.
The paper in the whole is well designed and results sound. Nevertheless, the manuscript needs a minor revision:
R1 #1 Abstract part should be re-written. It should be pointed more details of performed experiments.
In accordance with the reviewer’s suggestion, the abstract was revised and the results were brought in focus.
R1 #2 Figure 1 - should be more described in the text.
A detailed description of the information provided by Figure 1 is given in the revised manuscript.
R1 #3 Introduction part - the aim of the study should be pointed more.
The current methods for the quantification of “difficult” analytes are discussed in the revised manuscript to point out the possibilities of the novel methodology.
R1 #4 Figures 2 and 3 - should be removed to the experimental part.
Figure 2 provides the sequence of the labeled peptide; it was placed in the experimental part (now Figure 8). Figure 3 is an illustration of the analytical method. As this figure helps the reader to understand the strategy, it is of advantage to keep it at an early position of the manuscript. The authors would therefore suggest to keep it in its original place.
R1 #5 What about of Bioethical Comm. in case of real samples?
The blood samples were obtained from voluntary donors, this was stated in the revised manuscript.
Reviewer 2 Report
The manuscript describes the development of a new analytical method to quantify the presence of Pep19-2.5 in animal biological extract and Human serum. The authors describe a new method based on radioactively labeled derivatives. The results shown put in evidence as the developed method is an innovative tool that could improve the use of Pep19-2.5 in a correct way against sepsis. Furthermore, the developed method could be applied at other peptides that can't be extracted from biological samples.
I think that the manuscript could be accepted for publication, and it needs only a few minor revisions about the English language and style.
Author Response
Reviewer #2: The manuscript describes the development of a new analytical method to quantify the presence of Pep19-2.5 in animal biological extract and Human serum. The authors describe a new method based on radioactively labeled derivatives. The results shown put in evidence as the developed method is an innovative tool that could improve the use of Pep19-2.5 in a correct way against sepsis. Furthermore, the developed method could be applied at other peptides that can't be extracted from biological samples.
Thank you for your positive assessment of the novel analytical procedure.
I think that the manuscript could be accepted for publication, and it needs only a few minor revisions about the English language and style.